# Exploring Virulence Characteristics of Clinical *Escherichia coli* Isolates from Greece

**DOI:** 10.3390/microorganisms13071488

**Published:** 2025-06-26

**Authors:** Lazaros A. Gagaletsios, Elisavet Kikidou, Christos Galbenis, Ibrahim Bitar, Costas C. Papagiannitsis

**Affiliations:** 1Department of Microbiology, University Hospital of Larissa, 41334 Larissa, Greece; 2Department of Microbiology, Faculty of Medicine, University Hospital in Plzen, Charles University, 32300 Plzen, Czech Republic

**Keywords:** biofilm formation, *Escherichia coli*, CRISPR/Cas, plasmids, multidrug resistance

## Abstract

The aim of this study was to examine the genetic characteristics that could be associated with the virulence characteristics of *Escherichia coli* collected from clinical samples. A collection of 100 non-repetitive *E. coli* isolates was analyzed. All isolates were typed by MLST. String production, biofilm formation and serum resistance were examined for all isolates. Twenty *E. coli* isolates were completely sequenced Illumina platform. The results showed that the majority of *E. coli* isolates (87%) produced significant levels of biofilm, while none of the isolates were positive for string test and resistance to serum. Additionally, the presence of CRISPR/Cas systems (type I-E or I-F) was found in 18% of the isolates. Analysis of WGS data found that all sequenced isolates harbored a variety of virulence genes that could be implicated in adherence, invasion, iron uptake. Also, WGS data confirmed the presence of a wide variety of resistance genes, including ESBL- and carbapenemase-encoding genes. In conclusion, an important percentage (87%) of the *E. coli* isolates had a significant ability to form biofilm. Biofilms, due to their heterogeneous nature and ability to make microorganisms tolerant to multiple antimicrobials, complicate treatment strategies. Thus, in combination with the presence of multidrug resistance, expression of virulence factors could challenge antimicrobial therapy of infections caused by such bacteria.

## 1. Introduction

*Escherichia coli* is a Gram-negative bacterium colonizing the intestines of various animals including humans. Despite *E. coli* usually being a harmless bacterium, some isolates are pathogenic. *E. coli* can be the etiological agent of intestinal and extra-intestinal diseases, such as urinary tract infections, septicemia, peritonitis and pneumonia [1]. Annually, pathogenic *E. coli* (PEC) is estimated to cause over 2 million deaths globally [2]. PECs mainly include the enteropathogenic *E. coli* (EPEC), the Shiga toxin–producing *E. coli* (STEC) and the enterotoxigenic *E. coli* (ETEC) [3]. Additionally, among PECs, uropathogenic *E. coli* (UPEC) is responsible for up to 50% of hospital-acquired UTIs and for 90% of community UTIs in both men and women [4].

Pathogenicity is expedited by virulence factors like adhesion, iron acquisition, hemolysin, aerobactin’s and serum resistance encoded by genes found on plasmids and/or chromosome [5]. Furthermore, Shiga toxin (Stx) is considered a major virulence factor, associated with STEC infections [6]. The EPEC pathotype harbors the intimin (*eae*) gene [7], while the ETEC isolates harbor the plasmid-encoded heat-labile LT or heat-stable ST enterotoxins [8]. UPEC isolates have various virulence factors, including fimbriae and adhesins, biofilm formation ability, iron-acquisition factors, flagella, and toxins such as hemolysin [9,10]. These virulence factors provide the pathogen with the potential to evade or overwhelm host defense mechanisms, invade host cells, and induce inflammation in the host [11].

Furthermore, the growing problem of antimicrobial-resistance (AMR), associated with the presence of multiple resistance genes, complicates the antimicrobial treatment of infections. *E. coli* isolates have been reported to produce ESBLs, carbapenemases and/or MCRs conferring resistance to cephalosporins, carbapenems and polymyxins [12,13,14], respectively. These resistance genes are usually localized on mobile genetic elements, like plasmids, carrying additional resistance genes conferring resistance to multiple antimicrobial categories [15], such as aminoglycosides, fluoroquinolones, etc. Thus, treatment options of infections caused by these bacteria are limited to few choices with unpredictable effect [16].

Another interesting feature found in almost 40% of bacterial species is CRISPR/Cas (clustered regularly interspaced short palindromic repeats/CRISPR-associated genes or proteins) systems [17]. CRISPR/Cas systems, which were firstly detected on the *E. coli* K-12 chromosome, in 1987, are a defense mechanism against foreign invaders such as plasmids and viruses [18]. In addition, further studies have demonstrated that many Cas proteins from diverse CRISPR/Cas systems and organisms are involved in various functions, including gene regulation [19,20], DNA repair [21], and cell dormancy [22]. Interestingly, in 2009, CRISPR/Cas systems were shown to affect biofilm formation and swarming in *P. aeruginosa* [23,24].

Therefore, the aim of the current study was to examine the genetic characteristics that could be associated with virulence characteristics of *E. coli* isolates collected from clinical samples.

## 2. Materials and Methods

### 2.1. Clinical Isolates, Identification and Susceptibility Testing

A collection of 100 nonrepetitive *E. coli*, isolated from January 2020 to October 2020, from patients treated in UHL, which is a 650-bed tertiary care hospital in Thessaly (Central Greece), serving a population of 1,000,000 inhabitants, was studied. For each month, the ten first isolates were selected. Identification and antimicrobial susceptibility testing of bacterial isolates was performed by the VITEK-2 system (bioMérieux, Marcy l’Étoile, France). *E. coli* ATCC25922 and *Pseudomonas aeruginosa* ATCC 27853 were used as control strains [25]. Data were interpreted according to the criteria (version 15.0) of the European Committee on Antimicrobial Susceptibility Testing (EUCAST) (www.eucast.org, accessed on 1 February 2025).

### 2.2. Detection of ESBL-Encoding Genes

All isolates were screened, by PCR amplification, for the presence of genes encoding SHV and CTX-M β-lactamases, using specific primers [12].

### 2.3. Multilocus Sequence Typing

All *E. coli* isolates were typed by multilocus sequence typing (MLST), as described previously [26]. Sequence types (STs) and clonal complexes (CCs) were assigned at the *E. coli* MLST database (https://pubmlst.org/organisms/escherichia-spp, accessed on 1 February 2025).

### 2.4. String Test

All *E. coli* isolates were grown on blood agar plates overnight at 37 °C and stretched using an inoculation loop. The string was manually measured with a ruler. Isolates that produced colonies that could be stretched into a viscous string of >5 mm in all cases were considered string test positive.

### 2.5. Hemolysin Production

The ability of *E. coli* to induce hemolysis on blood agar was evaluated to detect the hemolysin-producing isolates. The isolates were inoculated into 5% sheep blood agar plates and incubated overnight at 37 °C. Hemolysin production was detected by the presence of a complete clearing zone of the erythrocytes around the colonies.

### 2.6. Microtiter Plate (MTP) Assay

The biofilm formation of *E. coli* was quantitatively estimated using the MTP assay as described earlier with minor modifications [27]. Briefly, 1% overnight culture of the test isolate was dispensed into 96-well MTP containing 200 μL of tryptone soya broth (TSB). The plate was incubated without agitation, for 24 h at 37 °C, to facilitate the adherence of the bacteria cells onto the surface of the MTP. Following incubation, the broth medium was discarded and the wells were gently washed twice with PBS to remove loosely attached planktonic cells. Each well was stained with 200 μL of 0.4% crystal violet (CV) solution for 1 min and the excess stain was removed by rinsing the wells thrice with sterile distilled water. The CV bound to the biofilm mass was then solubilized with 200 μL of 95% ethanol and quantified. Then, the absorbance was measured at 650 nm. The results were interpreted based on the recommendations published previously [28]. Specifically, isolates were classified as non-biofilm producers with OD650 values < 0.1, weak biofilm producers with OD650 = 0.1–0.2, moderate biofilm producers with OD650 = 0.2–0.4, and strong biofilm producers with OD650 values > 0.4.

### 2.7. Serum Resistance Assay

The recalcification method, as described previously [29,30], was used to prepare pooled human serum from at least 5 different people. Isolates were grown in BHI medium overnight at 37 °C with shaking. The isolates were spun down, washed two times with PBS, and adjusted to a final concentration of 1 × 10^7^ CFU/mL with PBS. A total of 1 × 10^5^ CFU was added to 5 mL of 80% NHS (normal human serum) in PBS and 80% heat-inactivated NHS in PBS. The samples were incubated with shaking at 37 °C for 0 and 3 h. After incubation, the cell count was determined using 10-fold serial dilutions and conventional plating in Luria Bertani agar. The log values of CFU counts at time 3 hrs normalized to CFU counts at time zero were calculated. The interpretation of the results was carried out based on the recommendation shown below. Isolates with log values < −0.7 were serum-sensitive, isolates with log values = −0.7–0.3 exhibited intermediate resistance to serum, and isolates with log values > 0.3 were serum-resistant.

### 2.8. CRISPR/Cas Typing and CRISPR Amplification

To perform CRISPR/Cas typing, we used the *cas2* gene to examine the presence of type I-E systems, and the I-F specific *cas1* gene to examine the presence of type I-F systems. Then, for all *E. coli* isolates, CRISPR arrays were PCR-amplified using specific primers. Primers were designed using WGS data of *E. coli*, harboring type I-E or I-F CRISPR/Cas systems, which were downloaded from the CRISPRCasdb database (https://crisprcas.i2bc.paris-saclay.fr/MainDb/StrainList, accessed on 1 February 2025). Primers and references are shown in Table 1. PCR amplicons were sequenced on both strands using an ABI 3500 sequencer (Applied Biosystems, Foster City, CA, USA). CRISPRFinder was chosen to confirm the CRISPR sequences obtained. Furthermore, the BLAST algorithm (www.ncbi.nlm.nih.gov/BLAST, accessed on 1 February 2025) was employed for the sequence analysis of the spacers identified.

### 2.9. Whole-Genome Sequencing

Twenty *E. coli* isolates, selected based on different susceptibility profiles, STs, virulence profiles and epidemiological data, were further analyzed with whole-genome sequencing (WGS). The genomic DNAs of the *E. coli* isolates, extracted using the DNA-Sorb-B kit (Sacace Biotechnologies S.r.l., Como, Italy), were sequenced using the Illumina MiSeq platform (Illumina Inc., San Diego, CA, USA). Initial paired-end reads, which were quality trimmed using Trimmomatic tool v0.32 (22 January 2022), were assembled via de Bruijn graph-based de novo assembler SPAdes v3.14.0 (31 December 2019).

### 2.10. Analysis of WGS Data

Antibiotic resistance genes were identified using the ResFinder 4.6 tool (http://genepi.food.dtu.dk/resfinder, accessed on 1 February 2025) with a threshold for minimum identity of 90% and for minimum coverage of 60% [32]. PlasmidFinder 2.1 (https://cge.food.dtu.dk/services/PlasmidFinder/, accessed on 1 February 2025), using a threshold for minimum identity of 95% and for minimum coverage of 60%, was used for detecting plasmid replicons in the sequenced isolates [33]. Additionally, WGS data were analyzed with VirulenceFinder 2.0 (https://cge.food.dtu.dk/services/VirulenceFinder/, accessed on 1 February 2025), with a threshold for minimum identity of 90% and for minimum coverage of 60% [34], to detect virulence factors that could be involved in the pathogenicity of the isolates. The SerotypeFinder 2.0 tool (https://cge.food.dtu.dk/services/SerotypeFinder/, accessed on 1 February 2025) with a threshold for minimum identity of 85% and for minimum coverage of 60% was also used to identify the serotype of *E. coli* [35].

### 2.11. Nucleotide Sequence Accession Numbers

Whole-genome assemblies of *E. coli* were deposited in NCBI under accession number PRJNA1257349.

## 3. Results

### 3.1. Metadata and Susceptibility Info

Among our collection, the majority of *E. coli* isolates were recovered from urine samples (*n* = 65), while 29 isolates were collected from blood samples. The remaining six *E. coli* were isolated from pus (*n* = 3), swab (*n* = 1) and sputum (*n* = 1) samples, while one isolate was of unknown source. Most of the samples were collected from the emergency medicine department (*n* = 45), indicating a community origin of the samples. Nevertheless, other samples were collected from different departments, including internal medicine (*n* = 28), urology (*n* = 8), pediatric (*n* = 5), surgery (*n* = 4), neurology, ICU (*n* = 3), outpatient (*n* = 2), orthopedics (*n* = 1), rheumatology (*n* = 1) and endocrinology (*n* = 1) (Appendix A).

Additionally, the clonal relatedness of the *E. coli* isolates, which was studied by MLST, showed the presence of 39 different STs among our collection. However, a significant number of isolates belonged to clonal complex 131 (CC131) (*n* = 45) (Table 2), including sequence types (STs) 131 (*n* = 30), 1195 (*n* = 1), 7399 (*n* = 7), 7597 (*n* = 5) and 10,605 (*n* = 2). Most of CC131 isolates were recovered from urine (*n* = 26) and blood (*n* = 14) samples. Thirty-two isolates were assigned to STs 69 (*n* = 2), 104 (*n* = 5), 390 (*n* = 2), 410 (*n* = 2), 476 (*n* = 2), 501 (*n* = 2), 569 (*n* = 2), 646 (*n* = 4), 922 (*n* = 3), 4560 (*n* = 3) and 9612 (*n* = 4), whereas the remaining 23 isolates belonged to unique STs (Appendix A). An analysis, using the eBURST algorithm [36], showed that most of the detected STs were not related to each other (Appendix A), while it confirmed that ST195, ST7399, ST7597 and ST10605 belonged to CC31.

Antimicrobial susceptibility testing showed that the frequency of resistance to ampicillin was 77%, while 61% of the *E. coli* isolates were resistant to amoxicillin/clavulanic acid, 38% were resistant to cefepime, 35% were resistant to ceftazidime, 27% were resistant to piperacillin/tazobactam, and 12% were resistant to carbapenems. Additionally, 51% of the *E. coli* isolates were resistant to ciprofloxacin, 45% were resistant to co-trimoxazole, 21% of the isolates were resistant gentamicin, and only 14% of the isolates were resistant to amikacin. Based on their susceptibility profiles, 44 isolates were classified as MDR (multi-drug resistance). PCR screening showed that 15 *E. coli* isolates carried genes encoding enzymes of CTX-M-1 family. In addition, eleven isolates carried genes encoding enzymes of CTX-M-9 family, while twelve other isolates were positive for both *bla*_CTX-M-1_-like and *bla*_CTX-M-9_-like genes. The majority of *bla*_CTX-M_-positive isolates belonged to CC131 (*n* = 27).

### 3.2. Virulence Characteristics

In regard to virulence characteristics, none of the isolates were positive for string test. Similarly, hemolysin production was observed only in 9% of the *E. coli* isolates. Isolates showing hemolysin production belonged to diverse STs and were recovered from urine (*n* = 6) and blood samples (*n* = 3) (Appendix A). The remaining isolates showed no hemolysis. From the biofilm quantification assay, we observed that the majority (*n* = 71) of *E. coli* isolates produced moderate levels of biofilm (0.4 > OD > 0.2). High production of biofilm was quantitatively estimated for 16 *E. coli* isolates, while the remaining 13 isolates produced weak levels of biofilm (0.2 > OD > 0.1). The majority of ST131 isolates (25 out of 30) produced moderate levels of biofilm. Isolates with high production of biofilm were recovered from urine (*n* = 8), blood (*n* = 6), pus (*n* = 1) and swab (*n* = 1) samples. Of note, all ST922 (*n* = 3) isolates showed high production of biofilm. Moreover, the serum-killing assay showed that no *E. coli* isolates were resistant to serum. The majority (91%) of *E. coli* isolates were serum-sensitive. The nine remaining *E. coli* exhibited intermediate levels of resistance to serum (Appendix A).

### 3.3. CRISPR/Cas Systems

Among 100 clinical *E. coli* isolates, CRISPR/Cas systems were detected in 18 isolates. Type I-E systems were found in 12 isolates. Isolates with I-E systems belonged to distinct STs (ST58, ST69, ST156, ST216, ST501, ST648, ST708, ST744, ST1011, ST9312) (Table 3). Two of the ST69 isolates harbored identical CRISPR arrays (Figure 1). Also, six isolates harbored a type I-F system. Three isolates carrying I-F systems were assigned to CC95, including STs 95 (*n* = 1) and 390 (*n* = 2) (Table 3). The three remaining type I-F-positive isolates were distributed in STs 569, 2371 and 5328. Of note was that none of the isolates, even of the same ST, harbored identical CRISPR arrays (Figure 1), indicating an ongoing evolution of the CRISPR/Cas systems. None of the CRISPR/Cas-positive isolates were found to be positive for more than one system.

For all CRISPR/Cas-positive isolates, CRISPR arrays were PCR-amplified and sequenced. In addition, type I-F CRISPR loci were amplified from one cas-negative isolate. These loci were determined as orphan CRISPR arrays. Unique spacers were named (Appendix A) and the arrangement of the spacers in each strain is shown in Figure 1. In isolates carrying type I-E systems, the CRISPR1 array ranged from 5 to 23 spacers, while the CRISPR2 array carried from 4 to 16 spacers. Of note, in two isolates (Eco-2194 [ST501] and Eco-2258 [ST708]), the CRISPR2 array was not amplified. On the other hand, in type I-F-positive isolates, the CRISPR1 array carried from 4 to 14 spacers, while the CRISPR2 array comprised from 4 to 14 spacers. Furthermore, among our collection, 290 unique spacers were found, with 212 spacers originating from type I-E systems, and 78 of them originating from type I-F systems. In order to determine the origin of the sequences of our spacers, a BLASTn analysis of spacers was carried out. Only results with high identity scores (100% coverage, ≥90% identity) were considered. A total of 257 (88.6%) spacers matched *E. coli* chromosomal sequences (*n* = 38) or CRISPR sequences (*n* = 219). Twelve (4.2%) spacers exhibited no significant similarity with sequences submitted to the NCBI database, which was expected, as a high number of spacers are still of unknown origin. The BLASTn search showed that 18 (6.2%) spacers matched to plasmids like pAVS0973-C (GenBank accession No. CP124474), pF7386-2 (GenBank accession No. CP038361) and p24C171-1 (GenBank accession No. LC501671) that were isolated from *E. coli* strains. Four of the later spacers were identical to regions involved in conjugative transfer, while another spacer was identical to a region implicated in plasmid replication. None of the identified plasmids carried antimicrobial resistance genes. We also identified three (1.0%) spacers that matched bacteriophages like isolates 3509_17389 (GenBank accession No. OP075566), 3121_76502 (GenBank accession No. OP075223) and vB_EcoM-813R1 (GenBank accession No. ON470617) (Appendix A).

### 3.4. WGS Data

For all isolates, Illumina sequencing resulted in sequences with a Pred quality score of >20. Additionally, following assembly, the obtained genomes were composed of contigs with a length-weighted coverage ranging from 18.5x to 88.1x, and N50 statistics ranging from 2165-bp to 30215-bp. An analysis of WGS data with ResFinder confirmed the presence of ESBL-encoding genes, *bla*_CTX-M-15_, *bla*_CTX-M-14_ and *bla*_CTX-M-27_, in eight of the sequenced isolates (Table 4). Additionally, WGS data showed that one ST9312 isolate carried the *bla*_NDM-1_ metallo-β-lactamase-encoding gene, and one ST648 isolate carried the *bla*_KPC-2_ crabapenemase-encoding gene, while one ST156 isolate harbored the colistin-resistance gene, *mcr1.1*. Furthermore, the majority of the isolates exhibited additional genes for resistance to aminoglycosides, chloramphenicol, trimethoprim, sulfonamides, tetracyclines, rifampicin, macrolides and quinolones. However, five of the sequenced *E. coli* isolates carried no resistance genes.

An analysis of WGS data with VirulenceFinder showed the presence of several different virulence genes in all isolates (Table 4). Different combinations of virulence genes were found even in isolates belonging to the same ST (Figure 2). However, the virulence genes *nlpI* encoding lipoprotein NlpI, *csgA* encoding the maor curlin subunit formed during biofilm formation, *terC* involved in tellurium ion resistance, and *yehC* associated with YHD fimbriael cluster were detected in all isolates. A significant number of isolates was positive for genes encoding adhesins (*fimH* [*n* = 18], *fdeC* [*n* = 15], and *papC* [*n* = 4]) [37]. The genes *kpsE* and *kpsM*, involved in biosynthesis of capsular polysaccharides, were detected in 12 isolates. Six isolates were positive for *lpfA* encoding long polar fimbriae, an adherence factor. The *mcbA* gene encoding a predicted periplasmic protein-encoding gene, YbiM, involved in the regulation of biofilm formation, was found in the ST10 isolate (Eco-7827). Interestingly, 16 isolates were positive for the *iss* gene associated with increased serum survival [38]. Several genes, like *gad* (*n* = 5; glutamate decarboxylase, which synthesizes Gamma-aminobutyric acid [GABA] from l-glutamic acid), *neuC* (*n* = 3; N-Acetylneuraminic acid [NeuAc] synthetic pathway) and *sat* (*n* = 4; encoding the serine acetyltransferase) involved in various metabolic pathways were also detected. Additionally, genes were found among the sequenced isolates, such as *tsh* (*n* = 6; encoding a temperature-sensitive hemagglutinin), *dnaK* (*n* = 3; encoding DnaK, also known as HSP70, that belongs to the superfamily of Heat shock proteins) and *clpK1* (*n* = 1; encoding a heat-resistant plasmid-borne ATPase), involved in the synthesis of stress-specific proteins [39]. Sequenced isolates also carried genes (*iutA* [*n* = 12], *ireA* [*n* = 4], *iucC* [*n* = 12], *iroN* [*n* = 11], and *aer* [*n* = 1]) encoding siderophore systems. In addition, sixteen isolates harbored the *sitA* gene encoding the ABC cassette importer SitABCD/MntABC for bacterial Mn homeostasis systems. The genes *fyuA* (*n* = 13; pyelonephritis-associated pilus [P fimbriae or pap] Yersinia bactin receptor), *vat* (*n* = 3; vacuolating autotransporter toxin), *chuA* (*n* = 14; heme receptor) and *yfcV* (*n* = 8; encoding the major subunit of a putative chaperone-usher fimbria), which are associated with UPEC isolates [40], were also found. The *estC* gene encoding an esterace that could be implicated in the ethyl acetate and/or ethyl lactate biosynthesis was detected in nine isolates. Twelve isolates carried genes encoding the HlyE (Hemolysin/cytolysin A) toxin. Also, the *hylF* encoding hemolysin F was found in 10 isolates. Four isolates were positive for gene *tia* encoding a determinant invasion previously described in ETEC isolates [41], while two isolates were positive for *astA* coding an Enteroaggregative heat-stable enterotoxin [42]. Additionally, three isolates carried genes encoding the Usp (Colicin-like Usp) toxin. Genes encoding colicins V (CvaV) and E4 (ColE4) were also found in nine and one isolates, respectively. Two isolates (Eco-3096 and Eco-6440) included a gene expressing *SenB* Enterotoxin. The three ST131 isolates carried the genes *ibeA* (an invasion actor of brain endothelium) and *iha* (which is associated with the Locus of Adhesion and Autoaggregation). Of note was that the ST501 isolate (Eco-2194) carried the *aap* and *aatA* genes, and the *agg* operon associated with EAEC pathotype [43].

Furthermore, an analysis of WGS data with PlasmidFinder detected a huge variety of plasmid replicons among the sequenced isolates (Table 4), with most of the replicons belonging to the IncF-type. However, in an ST69 *E. coli* isolate, which carried no resistance genes, no plasmid replicons were found. Despite the use of short-read sequencing technology, an analysis of WGS data found an association of the IncFIB (AP001918) replicon with the *hlyF* and *estC* virulence genes.

## 4. Discussion

In the current study, we examined the virulence characteristics of clinical *E. coli* isolates. The majority of the isolates were collected from urine and blood samples, indicating the infectivity and pathogenicity of the studied isolates. Our results showed that the majority (87%) of *E. coli* isolates had a significant ability to form biofilm. The biofilm EPS matrix protects bacteria from host immune responses and increases their resistance to antimicrobial agents, making infections difficult to treat. Biofilms can harbor persistent cells that enter an inactive state, which helps them overcome antibiotic treatments without genetic changes. These cells can re-establish active infections after the end of the treatment [44]. Furthermore, biofilms complicate treatment strategies due to their heterogeneous nature and ability to evade conventional therapeutic approaches [45,46].

A previous study has shown that 110 genes were associated with biofilm formation [47]. Those genes were involved in various functions, like for cell surface structures and cell membrane. The genes *fimH*, *csgA*, *nlpI* and *fdeC*, which were previously associated with biofilm formation [47], were observed in almost all *E. coli* isolates sequenced during this study (Figure 1). Of note, *dnaK*, which was also highlighted as an important gene for biofilm formation [47], was found in three isolates with moderate levels of biofilm formation. Another interesting observation was the fact that *yfcV* (a major subunit of a putative chaperone-usher fimbria), *fyuA* (P fimbriae), *chuA* (heme receptor), *lpfA* (long polar fimbriae), and genes involved in capsular biosynthesis (*kpsE* and *kpsM*) were mostly found among isolates that were not weak biofilm producers. Previous studies have shown that capsular polysaccharides also contribute to UPEC biofilm formation in the bladder [48,49].

On the other hand, none of the studied isolates formed a viscous string of >5 mm and were resistant to killing by complement. String phenotypes are associated with capsule production [30]. Complement is a critical part of the innate immune response, serving as a first line of defense to eliminate a diverse range of pathogens, including bacteria, viruses and parasites. In a previous study, 31.6% of *K. pneumoniae* isolates tested were resistant to killing by complement [50], which is in disagreement with the current data, indicating a diverse behavior among different species. However, only in nine isolates was hemolytic activity observed. Hemolysis, which is the lysis of red blood cells, is mainly caused by hemolysins, toxins that are produced by pathogenic bacteria. Isolate Eco-3092, which was positive for hemolysin production, carried the *hlyE* gene. On the other hand, two isolates (Eco-2253 and Eco-6371) carried the operon *hylABCD*-encoding Alpha-hemolysin but were negative for hemolysin production on blood agar. These contradictory results indicate that further experiments are needed to elucidate this mechanism.

Furthermore, an analysis of WGS data found that all sequenced isolates harbored a variety of virulence genes. A significant number of isolates was positive for genes (*fimH*, *fdeC*, and *papC*) for adhesion. Several genes, like *gad*, *neuC* and *sat*, involved in various metabolic pathways were also detected. Additionally, genes (*tsh*, *dnaK* and *clpK1*) involved in the synthesis of stress-specific proteins were found. Sequenced isolates also carried genes (*iutA*, *ireA*, *iucC*, *iroN*, and *aer*) encoding siderophore systems. Finally, *E. coli* isolates were positive for genes (*astA*, *usp*, *cvaC*, *colE4* and *senB*) encoding several toxins. These results highlight the huge armamentarium of virulence genes that can be involved in the pathogenicity of *E. coli* isolates. An association between the virulence genes and the STs or the virulence characteristics was not observed. However, the above findings show only the presence of the genes involved in the pathogenicity of *E. coli*. Discrepancies between virulence phenotypes and genotypes could be explained by frameshift mutations, truncations and/or incomplete loci.

In addition, our results showed that a significant number of *E. coli* isolates were resistant to β-lactam antibiotics (77% ampicillin, 35% cephalosporins, 12% carbapenems), complicating treatment strategies. Also, resistance to other antimicrobial classes, like aminoglycosides and fluroquinolones, was observed among the studied isolates. WGS data confirmed the presence of a wide variety of resistance genes, including ESBL-encoding genes, in most of the sequenced isolates. Interestingly, two isolates harbored a carbapenemase-encoding gene, *bla*_NDM-1_ or *bla*_KPC-2_, while the *mcr1.1* colistin-resistance gene was detected in an ST156 isolate. Although the majority of the *E. coli* isolates belonged to CC131, the wide variety of resistance genes that were observed could be explained by the fact that most of the isolates were of community origin.

Another interesting characteristic of our collection was the presence of CRISPR/Cas systems in 18% of the isolates. Type I-E systems were found in 12 isolates, and type I-F systems were present in 6 other isolates. None of the CRISPR/Cas-positive isolates were found to be positive for more than one system. Of note, the percentage of CRISPR/Cas-positive *E. coli* isolates was low compared to percentages observed in other bacterial species, like *P. aeruginosa* [51]. The low percentage of CRISPR/Cas-positive isolates might be associated with the wide variety of plasmid replicons observed in the sequenced isolates (Table 4). Several previous studies have reported that plasmids have been involved in the spread of resistance genes [52]. Also, several studies have demonstrated the critical role of plasmids in the dissemination of virulence factors [53,54]. Such examples were also observed during this study, referring to the association of IncFIB (AP001918) replicon with the *hlyF* and *estC* virulence genes. The *hlyF* encodes hemolysin F, a functional ortholog of CrpA, which is a short-chain dehydrogenase/reductase (SDR) that contributes to resistance against colistin and antimicrobial peptides. SDRs also induce the production of OMVs, which block autophagic flux. Previous studies have shown that HlyF is encoded by virulence plasmids of *E. coli* [53]. The *estC* gene encodes an esterase that could be implicated in the ethyl acetate and/or ethyl lactate biosynthesis. Previous studies have shown that the ester biosynthesis capacity by lactic acid bacteria (LAB) is of great interest in view of fruity flavor formation during sourdough and sourdough bread productions [55].

PCR amplification and sequencing of CRISPR loci identified 290 unique spacers. However, CRISPR loci were also amplified from one Cas-negative isolate. This finding may indicate that *cas* genes have been deleted through recombination events during the evolutionary history of bacteria. These orphan loci may provide info regarding the interaction history of the isolates with ‘invading molecules’ until the deletion event. Additionally, the presence of spacer sequences in these orphan loci may provide an adaptive immune memory to their hosts if they are found concurrently with complete CRISPR/Cas systems, enhancing their protection against infections by competitor phages [56]. The BLASTn analysis of unique spacers showed that the majority (88.6%) of them matched *E. coli* chromosomal sequences. Regarding the acquisition of self-targeting spacers, previous studies have proposed the potential role of the CRISPR/Cas system to regulate bacterial virulence. In *P. aeruginosa*, the CRISPR/Cas system enables modulation of biofilm formation, which is an important virulence factor for various pathogenic microorganisms [24]. A significant number of spacers (4.2%) exhibited no significant similarity with sequences submitted to the GenBank database, supposing that a significant number of genetic features have not yet been sequenced. The remaining 21 spacers matched to plasmids (*n* = 18) and phages (*n* = 3). However, none of the latter plasmids showed significant similarity with the plasmid sequences characterized during this study.

## 5. Conclusions

In conclusion, the analysis of virulence characteristics showed that an important percentage (87%) of *E. coli* isolates had a significant ability to form biofilm. Biofilms complicate treatment strategies due to their heterogeneous nature and ability to evade conventional therapeutic approaches. Also, *E. coli* isolates harbored a variety of virulence genes that could be implicated in adherence, invasion, and iron uptake. Despite further experiments being needed to elucidate the association of specific virulence genes with respective virulence phenotypes, the expression of several virulence factors, in combination with the presence of multidrug resistance, could challenge the antimicrobial therapy of infections caused by such bacteria.

## Figures and Tables

**Figure 1 microorganisms-13-01488-f001:**
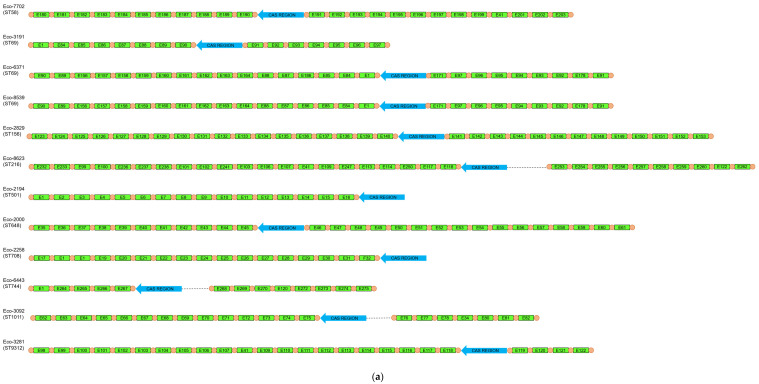
Schematic representation of the spacers’ arrangement in type I-E (**a**) and I-F (**b**) CRISPR/Cas-positive *E. coli* isolates, characterized during this study.

**Figure 2 microorganisms-13-01488-f002:**
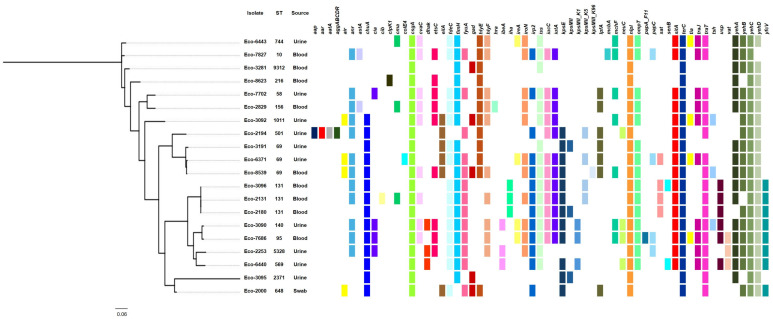
SNP-based phylogeny of the 20 *E. coli* isolates characterized by Illumina sequencing. The colored squares indicate the presence of selected virulence genes, which were detected by VirulenceFinder 2.0.

**Table 1 microorganisms-13-01488-t001:** Primers used for amplification in the present study.

Primers	Sequence (5′→3′)	References
C1Fw	GTTATGCGGATAATGCTACC	[31]
C1Rev	CGTAYYCCGGTRGATTTGGA	[31]
C2Fw	AAATCGTATGAAGTGATGCAT	[31]
C2Rev	GTCGATGCAAACACATAAATA	[31]
C3Fw	GCGCTGGATAAAGAGAAAAAT	[31]
C3Rev	GCCCACCATTCACCTGTA	[31]
C4Fw	CTGAACAGCGGACTGATTTA	[31]
C4Rev	GTACGACCTGAGCAAAG	[31]
Ec-Cas1-F	ATGTCTTCGAATTACCTTACG	This study
Ec-Cas1-R	TCATTGGTCAGCCTTAGCCA	This study
Ec-Cas2-F	ATGAGTATGGTGGTTGTGGTC	This study
Ec-Cas2-R	TTATTGATTTTCAACAGGAAGA	This study

**Table 2 microorganisms-13-01488-t002:** Epidemiological characteristics of 100 *E. coli* isolates included in this study.

CCs*/STs (Number of Isolates)	Percentage (%)	Source (Number of Isolates)	Department (Number of Isolates)
**CC10** ST10 (1) ST744 (1)	2%	Blood (1) Urine (1)	Emergency medicine (1) Urology (1)
ST14	1%	Urine (1)	Emergency medicine (1)
ST58	1%	Urine (1)	Emergency medicine (1)
**CC69** ST69 (3) ST922 (3)	6%	Urine (5) Blood (1)	Internal medicine (2) Neurology (2) Surgery (1) Pediatrics (1)
**CC95** ST95 (1) ST140 (1) ST390 (2)	4%	Urine (2) Blood (2)	Emergency medicine (3) Pediatrics (1)
ST104	5%	Urine (5)	Emergency medicine (3) Outpatient (1) Endocrinology (1)
**CC131** ST131 (30) ST1195 (1) ST7379 (7) ST7527 (5) ST10605 (2)	45%	Urine (27) Blood (14) Sputum (1) Pus (2) Unknown (1)	Emergency medicine (22) Internal medicine (14) Urology (4) ICU (2) Rheumatology (1) Surgery (2)
ST156	1%	Blood (1)	Internal medicine (1)
ST186	1%	Urine (1)	Pediatrics (1)
ST216	1%	Blood (1)	Internal medicine (1)
**CC410** ST410 (2) ST3059 (1)	3%	Urine (3)	Emergency medicine (1) Internal medicine (1) Pediatrics (1)
ST476	2%	Urine (1) Blood (1)	Emergency medicine 2%
ST501	2%	Urine (1) Blood (1)	Internal medicine (1) Outpatient (1)
ST569	2%	Urine (1) Blood (1)	Internal medicine (1) Emergency medicine (1)
ST646	4%	Urine (4)	Emergency medicine (3) Internal medicine (1)
ST648	1%	Swab (1)	Internal medicine (1)
ST708	1%	Urine (1)	Urology (1)
ST1011	1%	Urine (1)	Internal medicine (1)
ST1133	1%	Urine (1)	Internal medicine (1)
ST1432	1%	Urine	Emergency medicine
ST1538	1%	Pus (1)	Urology (1)
ST2371	1%	Urine (1)	Urology (1)
ST3387	1%	Urine (1)	Orthopedics (1)
ST3423	1%	Blood (1)	Pediatrics (1)
ST3459	1%	Urine (1)	Emergency medicine (1)
ST4077	1%	Urine (1)	Internal medicine (1)
ST4560	3%	Urine (1) Blood (2)	Emergency medicine (2) Neurology (1)
ST5328	1%	Urine (1)	Internal medicine (1)
ST9312	1%	Blood (1)	Emergency medicine (1)
ST9612	4%	Urine (2) Blood (2)	Emergency medicine (2) Internal medicine (1) Surgery (1)

* Clonal complexes. Clonal complexes are shown in bold format.

**Table 3 microorganisms-13-01488-t003:** Characteristics of CRISPR/Cas-positive *E. coli* isolates of Greek origin.

Isolate	Source	Department	ST	Resistance Genes	CRISPR/Cas System	CRISPR 1	CRISPR 2
Eco-7702	Urine	Emergency medicine	58	*bla*_TEM-1_, *aph(3″)-Ib*, *aph(6)-Id*, *dfrA5*, *sul2*	Type I-E	11 spacers	13 spacers
Eco-3191	Urine	Neurology	69	No	Type I-E	8 spacers	7 spacers
Eco-6371	Urine	Internal medicine	69	*bla*_TEM-1_, *aph(3″)-Ib*, *aph(6)-Id*, *dfrA7*, *sul1*, *sul2*, *tetA*	Type I-E	17 spacers	9 spacers
Eco-8539	Blood	Surgery	69	*bla*_TEM-1_, *aph(3″)-Ib*, *aph(6)-Id*, *dfrA14*, *sul2*, *tet(A)*, *qnrS1*	Type I-E	17 spacers	9 spacers
Eco-7686	Blood	Emergency medicine	95	No	Type I-F	5 spacers	4 spacers
Eco-2829	Blood	Internal medicine	156	*bla_TEM-1_*, *aac(6′)-Ib3*, *catB3*, *dfrA1*, *mcr1.1*, *sul1*, *tet(B)*	Type I-E	18 spacers	13 spacers
Eco-8623	Blood	Internal medicine	216	No	Type I-E	21 spacers	10 spacers
Eco-7461	Urine	Emergency medicine	390	No *	Type I-F	5 spacers	6 spacers
Eco-8617	Blood	Pediatrics	390	No *	Type I-F	8 spacers	6 spacers
Eco-2194	Urine	Outpatient	501	*bla*_CTX-M-15_, *aph(3″)-Ib*, *aph(6)-Id*, *mph(A)*, *sul2*	Type I-E	16 spacers	-
Eco-6440	Urine	Internal medicine	569	*bla* _TEM-1_	Type I-F	13 spacers	7 spacers
Eco-2000	Swab	Internal medicine	648	*bla*_KPC-2_, *bla*_VEB-1_, *bla*_OXA-10_, *bla*_OXA-1_*. aac(6′)-Ib-cr*, *aadA1*, *aadB*, *aph(3″)-Ib*, *aph(6)-Id*, *arr-2*, *catB3*, *cmlA1*, *dfrA14*, *dfrA23*, *sul2*, *tet(A)*	Type I-E	11 spacers	16 spacers
Eco-2258	Urine	Urology	708	No *	Type I-E	17 spacers	-
Eco-6443	Urine	Urology	744	*bla*_TEM-1_, *aac(6′)-Ib*, *aph(3″)-Ib*, *aph(6)-Id*, *aadA2*, *aadA5*, *dfrA17*, *sul1*, *tet(B)*	Type I-E	5 spacers	8 spacers
Eco-3092	Urine	Internal medicine	1011	*bla*_CTX-M-14_, *aac(3)-IId*, *aadA1*, *aadA2*, *catA1*, *dfrA1*, *mph(A)*, *sul*, *tet(A)*	Type I-E	14 spacers	7 spacers
Eco-3095	Urine	Urology	2371	*bla*_CTX-M-14_, *bla*_TEM-1_	Type I-F	14 spacers	13 spacers
Eco-2253	Urine	Internal medicine	5328	No	Type I-F	10 spacers	6 spacers
Eco-3281	Blood	Emergency medicine	9312	*bla*_NDM-1_, *bla*_CTX-M-15_, *bla*_OXA-1_, *bla*_TEM-1_, *aac(6′)-Ib*, *aph(3″)-Ib*, *aph(6)-Id*, *catB3*, *dfrA14*, *sul2*, *tet(A)*	Type I-E	21 spacers	4 spacers

* No WGS data.

**Table 4 microorganisms-13-01488-t004:** Characteristics of 20 *E. coli* isolates characterized by Illumina sequencing.

Isolate	ST	Serotype	Resistance Genes	Plasmid Replicons	Virulence Genes	CRISPR/Cas System
Eco-7827	10	O8: H17	No	IncFIB (AP001918), IncFIC (FII), IncFII (29)	*astA*, *csgA*, *cvaC*, *etsC*, *fimH*, *fyuA*, *hlyEF*, *iroN*, *irp2*, *iss*, *iucC*, *iutA*, *mcbA*, *mchF*, *nlpI*, *ompT*, *papC*, *terC*, *traJ*, *traT*, *yehABC*	-
Eco-7702	58	O25: H8	*bla*_TEM-1_, *aph(3″)-Ib*, *aph(6)-Id*, *dfrA5*, *sul2*	IncFIB (AP001918), IncFII, IncQ	*cia*, *csgA*, *cvaC*, *etsC*, *fdeC*, *fimH*, *fyuA*, *hlyEF*, *iroN*, *irp2*, *iss*, *iucC*, *iutA*, *lpfA*, *mchF*, *nlpI*, *ompT*, *sitA*, *terC*, *traJ*, *traT*, *yehABCD*	Type I-E
Eco-3191	69	O17: H18	No	No	*chuA*, *csgA*, *eilA*, *fdeC*, *fimH*, *hlyE*, *iss*, *kpsE*, *kpsMII*, *lpfA*, *nlpI*, *ompT*, *sitA*, *terC*, *yehABCD*	Type I-E
Eco-6371	69	O15: H18	*bla*_TEM-1_, *aph(3″)-Ib*, *aph(6)-Id*, *dfrA7*, *sul1*, *sul2*, *tetA*	IncFIA, IncFIB (AP001918), IncFII, IncQ1, Col156, Col440I	*air*, *anr*, *chuA*, *colE4*, *csgA*, *eilA*, *fdeC*, *fimH*, *fyuA*, *hylAE*, *ireA*, *iroN*, *irp2*, *iss*, *iucC*, *iutA*, *kpsE*, *kpsMII*, *lpfA*, *nlpI*, *ompT*, *papC*, *sat*, *sitA*, *terC*, *tiA*, *traJ*, *traT*, *yehABCD*	Type I-E
Eco-8539	69	O15: H18	*bla*_TEM-1_, *aph(3″)-Ib*, *aph(6)-Id*, *dfrA14*, *sul2*, *tet(A)*, *qnrS1*	IncFIA, IncFIB (AP001918), IncFIC (FII)	*chuA*, *csgA*, *cvaC*, *eilA*, *etsC*, *fdeC*, *fimH*, *fyuA*, *hlyEF*, *iroN*, *irp2*, *iss*, *iucC*, *iutA*, *kpsE*, *kpsMII*, *lpfA*, *mchF*, *nlpI*, *ompT*, *sitA*, *terC*, *traT*, *tsh*, *yehABCD*	Type I-E
Eco-7686	95	O1: H7	No	IncFIB (AP001918), IncFII, Col8282	*chuA*, *cia*, *csgA*, *cvaC*, *etsC*, *fimH*, *fyuA*, *hlyF*, *ireA*, *iroN*, *irp2*, *iss*, *iucC*, *iutA*, *kpsE*, *kpsMII*, *mchF*, *neuC*, *nlpI*, *ompT*, *papA_F11*, *papC*, *sitA*, *terC*, *tia*, *traJ*, *traT*, *usp*, *yehABCD*, *yfcV*	Type I-F
Eco-2131	131	O25: H4	*bla*_CTX-M-15_, *bla*_OXA-1_, *bla*_TEM-1_, *aac(6′)-Ib-cr*, *aadA2*, *aph(3″)-Ib*, *catA1*, *dfrA12*, *dfrA14*, *mph(A)*, *sul1*, *sul2*, *tet(A)*	IncB, IncFIB (AP001918), IncFII (pCoo), ColpEC648	*chuA*, *cib*, *cma*, *cvaC*, *fimH*, *fyuA*, *hlyF*, *iha*, *iroN*, *irp2*, *iss*, *iucC*, *iutA*, *kpsE*, *kpsMII*, *nlpI*, *ompT*, *sat*, *terC*, *traT*, *yehACD*, *yfcV*	-
Eco-2180	131	O25: H4	*bla*_CTX-M-15_, *bla*_OXA-1_, *aac(6′)-Ib-cr*, *aadA5*, *catB3*, *dfrA17*, *mph(A)*, *sul1*	IncFIA, IncFII, IncI1-I (Alpha)	*afaACD*, *chuA*, *fimH*, *fyuA*, *iha*, *irp2*, *iss*, *iucC*, *iutA*, *kpsE*, *kpsMII*, *nlpI*, *ompT*, *sat*, *shiA*, *sitA*, *terC*, *traT*, *yehACD*, *yfcV*	-
Eco-3096	131	O25: H4	*bla*_CTX-M-27_, *aadA5*, *aph(3″)-Ib*, *aph(6)-Id*, *dfrA17*, *mph(A)*, *sul1*, *sul2*	IncFIA, IncFIB (AP001918), IncFII(pRSB107), Col156	*chuA*, *fimH*, *fyuA*, *iha*, *irp2*, *iss*, *iucC*, *iutA*, *kpsE*, *kpsMII*, *nlpI*, *ompT*, *sat*, *senB*, *sitA*, *terC*, *traT*, *yehACD*, *yfcV*	-
Eco-3090	140	O50: H5	*bla*_CTX-M-14_, *aac(6′)-Ib3*, *cmlA1*, *mph(A)*	IncFIB (AP001918), IncFIC (FII), IncFII	*chuA*, *csgA*, *cvaC*, *dnaK*, *etsC*, *fimH*, *fyuA*, *hlyF*, *ibeA*, *ireA*, *iroN*, *irp2*, *iss*, *iucC*, *iutA*, *kpsE*, *kpsMII*, *mchF*, *neuC*, *nlpI*, *ompT*, *sitA*, *terC*, *traJ*, *traT*, *usp*, *yehABCD*, *yfcV*	-
Eco-2829	156	O159: H28	*bla_TEM-1_*, *aac(6′)-Ib3*, *catB3*, *dfrA1*, *mcr1.1*, *sul1*, *tet(B)*	IncFIB (AP001918), IncFIC (FII), IncX4	*astA*, *cma*, *csgA*, *cvaC*, *etsC*, *fdeC*, *hlyEF*, *hra*, *iroN*, *iss*, *iucC*, *iutA*, *lpfA*, *nlpI*, *ompT*, *sitA*, *terC*, *traJ*, *traT*, *yehABCD*	Type IE
Eco-8623	216	O3: H4	*No*	IncY	*clpK1*, *csgA*, *etsC*, *fimH*, *hlyE*, *nlpI*, *terC*,	Type I-E
Eco-2194	501	O86: H4	*bla*_CTX-M-15_, *aph(3″)-Ib*, *aph(6)-Id*, *mph(A)*, *sul2*	IncFIB (AP001918), IncFII (pRSB107)	*aap*, *aar*, *aatA*, *aggABCDR*, *chuA*, *csgA*, *eilA*, *fyuA*, *hlyE*, *irp2*, *iucC*, *iutA*, *kpsE*, *kpsMII*, *lpfA*, *neuC*, *nlpI*, *sitA*, *terC*, *traT*, *yehBCD*	Type I-E
Eco-6440	569	O134: H31	*bla* _TEM-1_	IncFIB (AP001918), IncFII (29), Col156	*chuA*, *csgA*, *dhak*, *fimH*, *fyuA*, *ibeA*, *irp2*, *kpsE*, *kpsMII*, *neuC*, *nlpI*, *ompT*, *senB*, *sitA*, *terC*, *traJ*, *traT*, *usp*, *vat*, *yehABCD*, *yfcV*	Type I-F
Eco-2000	648	O45: H6	*bla*_KPC-2_, *bla*_VEB-1_, *bla*_OXA-10_, *bla*_OXA-1_*. aac(6′)-Ib-cr*, *aadA1*, *aadB*, *aph(3″)-Ib*, *aph(6)-Id*, *arr-2*, *catB3*, *cmlA1*, *dfrA14*, *dfrA23*, *sul2*, *tet(A)*	IncB, IncC, IncFIB (H89-PhagePlasmid), IncN, IncY	*chuA*, *csgA*, *eilA*, *fyuA*, *gad*, *hlyE*, *irp2*, *kpsE*, *lpfA*, *nlpI*, *terC*, *traT*, *yehBCD*, *yfcV*	Type I-E
Eco-6443	744	O101: H9	*bla*_TEM-1_, *aac(6′)-Ib*, *aph(3″)-Ib*, *aph(6)-Id*, *aadA2*, *aadA5*, *dfrA17*, *sul1*, *tet(B)*	IncFIB (AP001918), IncFIB (K), IncFII, IncFII (K), IncR	*cma*, *csgA*, *cvaC*, *himH*, *hlyEF*, *ireA*, *iroN*, *iss*, *mchF*, *nlpI*, *ompT*, *sitA*, *terC*, *tia*, *traJ*, *traT*, *yehABCD*	Type I-E
Eco-3092	1011	O102: H28	*bla*_CTX-M-14_, *aac(3)-IId*, *aadA1*, *aadA2*, *catA1*, *dfrA1*, *mph(A)*, *sul*, *tet(A)*	IncFIB (AP001918), IncFIC (FII), IncI1-I (Alpha)	*chuA*, *cvaC*, *eilA*, *etsC*, *fdeC*, *fimH*, *hlyEF*, *iroN*, *iss*, *iucC*, *iutA*, *mchF*, *nlpI*, *ompT*, *sitA*, *terC*, *tia*, *traJ*, *traT*, *tsh*, *yehBCD*	Type I-E
Eco-3095	2371	O1: H42	*bla*_CTX-M-14_, *bla*_TEM-1_	IncI1-I (Alpha)	*chuA*, *fimH*, *kpsE*, *kpsMII*, *nlpI*, *terC*, *yehCD*	Type I-F
Eco-2253	5328	O185: H10	No	IncFIB (AP001918), IncFIC (FII)	*chuA*, *cia*, *dhak*, *etsC*, *fimH*, *fyuA*, *hlyF*, *ibeA*, *iroN*, *irp2*, *iss*, *nlpI*, *ompT*, *sitA*, *terC*, *traJ*, *traT*, *vat*, *yehABCD*, *yfcV*	Type I-F
Eco-3281	9312	Not found	*bla*_NDM-1_, *bla*_CTX-M-15_, *bla*_OXA-1_, *bla*_TEM-1_, *aac(6′)-Ib*, *aph(3″)-Ib*, *aph(6)-Id*, *catB3*, *dfrA14*, *sul2*, *tet(A)*	IncC, p0111, repB (R1701)	*csgA*, *fimH*, *gad*, *hlyE*, *iss*, *nlpI*, *terC*, *yehABCD*	Type IE

## Data Availability

Whole-genome assemblies of *E. coli* were deposited in NCBI [https://www.ncbi.nlm.nih.gov/bioproject/?term=PRJNA1257349] (accessed on 30 April 2025) under accession number PRJNA1257349.

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
