# Peer review of "Exploring Virulence Characteristics of Clinical Escherichia coli Isolates from Greece"

_microorganisms, 2025, doi:10.3390/microorganisms13071488_

Round 1
Reviewer 1 Report
Comments and Suggestions for Authors
In the present, very interesting study, Gagaletstsios L et al aimed to provide insight into the virulence genetic determinants of clinical E.coli isolates from both community and hospital setting in central Greece. The authors have thoroughly studied the strain collection, both phenotypically and genotypically regarding their virulence characteristics and their antimicrobial susceptibility profiles using a well-structured study protocol combining classic microbiology assays and last-line molecular techniques.
The whole manuscript is well-written, the methods presented with details and the results are presented clearly both in text and in the 2 Tables & 2 Figures.
I would suggest a few minor changes to enhance clarity that would be helpful for the readers:
- Line 29, please correct “gram” to “Gram”
- The microorganism’s name should be written in italics throughout the manuscript. Please correct accordingly in lines 50,58,75, 248, 323, 325 as well as in the abstract.
- Line 102 correct “107” to “107”
- Line 157, I would suggest “Internal medicine department” instead of “pathology department”
- Line 158, I would suggest “outpatient department/emergency room” instead of “external clinics”
- Line 192, what do you mean by “3.2 Subsection”? You have used 3.2 for the Virulence characteristics results section
- Regarding to the evaluation of hemolysin production as a virulence factor, you present the results without having mentioned earlier in the methods section how you have studied this characteristic. Please add the relevant method.
- Could you specify if the 9 isolates you have found with intermediate level of serum resistance were recovered from a bloodstream infection? Accordingly, could you provide the source of the 16 isolates found with high production of biofilm? In general, it would be nice to add the clinical specimen source, at least in Table 2, for the isolates that have been studied with WGS regarding their virulence genes.
- Lines 254 and 257 regarding previous studies’ findings belong to the discussion and not the results section. Please modify accordingly.
- Lines 288-290 “Additionally, two isolates (Eco-2253 and Eco-6371) carried the operon (hylABCD) encoding Alpha-hemolysin, while two other isolates (Eco-3096 and Eco-6440) included a gene expressing SenB Enterotoxin” belong to the results’ section and not in the discussion.
- Since you have studied thoroughly the CRISPR/Cas systems found in 18 coli isolates and you report and discuss your findings with many details, I would suggest to add some information about those systems in the introduction, mainly how they could regulate genes associated with virulence beside the immunity they provide against mobile elements.
- I would suggest to add in the WGS results section a short description of the various virulence genes found in relation to their specific function, e.g. genes for adherence, genes for iron uptake or genes from several secretion systems etc
- Finally, since you have studied virulence characteristics of clinical E.coli isolates in general (meaning not only the genetic determinants but also the expression of specific virulence factors like biofilm and haemolysin production, serum resistance and hypermucoviscosity, I would suggest to modify your title from “Exploring Genetic characteristics associated with Virulence of clinical Escherichia coli isolates from Greece” to “Exploring virulence characteristics of clinical Escherichia coli isolates from Greece”.
Author Response
Commment 1: Line 29, please correct “gram” to “Gram”
RESPONSE 1: Done
Comment 2: The microorganism’s name should be written in italics throughout the manuscript. Please correct accordingly in lines 50,58,75, 248, 323, 325 as well as in the abstract.
RESPONSE 2: We apologize for the wrong format. In the revised manuscript, we microorganism’s names have been writhen in italics format.
Comment 3 : Line 102 correct “107” to “107”
RESPONSE 3: Done.
Comment 4: Line 157, I would suggest “Internal medicine department” instead of “pathology department”
RESPONSE 4: Done.
Comment 5: Line 158, I would suggest “outpatient department/emergency room” instead of “external clinics”
RESPONSE 5: Done
Comment 6: Line 192, what do you mean by “3.2 Subsection”? You have used 3.2 for the Virulence characteristics results section
RESPONSE 6: In the revised text, it has been corrected to 3.3 CRISPR/Cas systems.
Comment 7: Regarding to the evaluation of hemolysin production as a virulence factor, you present the results without having mentioned earlier in the methods section how you have studied this characteristic. Please add the relevant method.
RESPONSE 7: A description regarding the methodology for hemolysin production has been included in the revised text (Please see lines 93-97).
Comment 8: Could you specify if the 9 isolates you have found with intermediate level of serum resistance were recovered from a bloodstream infection? Accordingly, could you provide the source of the 16 isolates found with high production of biofilm? In general, it would be nice to add the clinical specimen source, at least in Table 2, for the isolates that have been studied with WGS regarding their virulence genes.
RESPONSE 8: Only 3 out 9 isolates with intermediate level of serum resistance were recovered from blood samples. Whereas, the isolates with high production of biofilm were recovered from urine (n=8), blood (n=6), swab (n=1) and pus (n=1) samples. This info has been mentioned in the revised text (please see lines 207-208 and 213-214).
Additionally, the clinical specimen source is mentioned in Table S1.
Comment 9: Lines 254 and 257 regarding previous studies’ findings belong to the discussion and not the results section. Please modify accordingly.
RESPONSE 9: As the Reviewer suggested, this part has been moved to the Discussion section, in the revised text (Please see lines 409-417).
Comment 10: Lines 288-290 “Additionally, two isolates (Eco-2253 and Eco-6371) carried the operon (hylABCD) encoding Alpha-hemolysin, while two other isolates (Eco-3096 and Eco-6440) included a gene expressing SenB Enterotoxin” belong to the results’ section and not in the discussion.
RESPONSE 10: As the Reviewer suggested, this sentence has been moved in the Results section, in the revised text (please see lines 310-311).
Comment 11: Since you have studied thoroughly the CRISPR/Cas systems found in 18 coli isolates and you report and discuss your findings with many details, I would suggest to add some information about those systems in the introduction, mainly how they could regulate genes associated with virulence beside the immunity they provide against mobile elements.
RESPONSE 11: In the revised manuscript, a small part discussing the role of CRISPR/Cas systems in the immunity against mobile elements and in the regulation of genes associated with virulence has been added (please see lines 58-66).
Comment 12: I would suggest to add in the WGS results section a short description of the various virulence genes found in relation to their specific function, e.g. genes for adherence, genes for iron uptake or genes from several secretion systems etc
RESPONSE 12: In the revised manuscript, a short description of the virulence genes has been added (Please see lines 281-315).
Comment 13: Finally, since you have studied virulence characteristics of clinical E.coli isolates in general (meaning not only the genetic determinants but also the expression of specific virulence factors like biofilm and haemolysin production, serum resistance and hypermucoviscosity, I would suggest to modify your title from “Exploring Genetic characteristics associated with Virulence of clinical Escherichia coli isolates from Greece” to “Exploring virulence characteristics of clinical Escherichia coli isolates from Greece”.
RESPONSE 13: In the revised manuscript, the title has been modified according to the Reviewer’s suggestion.
Reviewer 2 Report
Comments and Suggestions for Authors
Thank you for the opportunity to review the manuscript titled "Exploring Genetic Characteristics Associated with Virulence of Two Clinical Escherichia coli Isolates from Greece." This is an interesting study with a rich sampling. Although the novelty is somewhat limited, the data presented represent a valuable contribution to the literature. Congrats! I have some minor comments and suggestions.
Please review the entire manuscript to ensure that all scientific names (e.g., E. coli) are properly italicized. Additionally, there are several punctuation errors and typographical mistakes throughout the text. For example, in lines 64, 58, 102, and Table 1 (third column). A thorough language revision is recommended.
More information is needed regarding the sequencing quality. Please include basic metrics such as Phred score, coverage, and read length.
Lines 161-167: Percentages should be provided alongside the data. Additionally, a summary table showing the most frequent clonal complexes (CCs) and their respective percentages should be included in the main text, not only in the supplementary materials. This would greatly enhance the reader’s understanding of the findings.
Lines 161-178: Consider summarizing only the most relevant results in the main text. A full description of all findings is unnecessary when detailed tables and supplementary materials are already provided. Highlighting key trends and directing the reader to the most important patterns would be more effective.
Section 3.2: I recommend organizing this information into a table, at least summarizing the main antimicrobial resistance determinants and associated CC/STs. An eBURST diagram showing the relationships between the detected sequence types would also be a valuable visual addition.
Table 3: It may not be essential to include Table 3 in the main text, since most of its content is already covered in Figure 1-except for the resistance genes. Consider moving Table 3 to the supplementary material. Additionally, for Figure 1, it would be helpful to include the origin of each isolate (e.g., blood, urine, etc.).
The discussion section feels brief and somewhat superficial given the range and depth of the results. Expanding this section to provide a more thorough interpretation of key findings and their implications would strengthen the manuscript considerably.
Author Response
Comment 1: Please review the entire manuscript to ensure that all scientific names (e.g., E. coli) are properly italicized. Additionally, there are several punctuation errors and typographical mistakes throughout the text. For example, in lines 64, 58, 102, and Table 1 (third column). A thorough language revision is recommended.
RESPONSE 1: The text has been revised, species names have been written in italics format, and typographical errors have been corrected.
Comment 2: More information is needed regarding the sequencing quality. Please include basic metrics such as Phred score, coverage, and read length.
RESPONSE 2: In the revised manuscript, more info regarding the sequencing quality has been added. Please see lines 264-267.
Comment 3: Lines 161-167: Percentages should be provided alongside the data. Additionally, a summary table showing the most frequent clonal complexes (CCs) and their respective percentages should be included in the main text, not only in the supplementary materials. This would greatly enhance the reader’s understanding of the findings.
RESPONSE 3: Adding percentages next to the results doesn’t make sense, since the total amount of isolates was 100. Thus, the number of isolates is equal to the percentage.
However, a table with the percentages of different CCs/STs has been included in the revised manuscript (please, see Table 2 in the revised manuscript).
Comment 4: Lines 161-178: Consider summarizing only the most relevant results in the main text. A full description of all findings is unnecessary when detailed tables and supplementary materials are already provided. Highlighting key trends and directing the reader to the most important patterns would be more effective.
RESPONSE 4: This part of the of the text is already too synoptic, and there is not much repetition with the Tables and the Figures. Thus, we don’t understand of what findings the description is unnecessary.
Comment 5: Section 3.2: I recommend organizing this information into a table, at least summarizing the main antimicrobial resistance determinants and associated CC/STs. An eBURST diagram showing the relationships between the detected sequence types would also be a valuable visual addition.
RESPONSE 5: Since, we weren’t sure to which part of the text this comment refers, we have updated Table S1. Also, we have added a Table summarizing the characteristics of CRISPR/Cas-positive isolates in the revised manuscript (please see Table 3).
Additionally, as the Reviewer suggested, we have included an e BURST diagram showing the relations of the detected STS of the E. coli isolates studied during this study (please see Figure S1).
Comment 6: Table 3: It may not be essential to include Table 3 in the main text, since most of its content is already covered in Figure 1-except for the resistance genes. Consider moving Table 3 to the supplementary material. Additionally, for Figure 1, it would be helpful to include the origin of each isolate (e.g., blood, urine, etc.).
RESPONSE 6: We strongly believe that Table 3 should stay in the main text, since it summarizes the main results of WGS data. Replicons, resistance genes, serotype, and CRISPR/Cas systems are not shown in Figure 1.
However, as the Reviewer suggested, the clinical origin of each isolate has been included in Figure 1.
Comment 7: The discussion section feels brief and somewhat superficial given the range and depth of the results. Expanding this section to provide a more thorough interpretation of key findings and their implications would strengthen the manuscript considerably.
RESPONSE 7: In the revised manuscript, the discussion has been updated.
Reviewer 3 Report
Comments and Suggestions for Authors
Dear authors,
Congratulations on the excellent work, I consider the molecular study of clinical isolates of E. coli to be very advanced, but the points below need to be resolved:
- The abstract and introduction are adequate. The introduction provides current information on the subject.
- Lines 84-85 cite the classic reference, in this case Stepanovic 2007 (10.1111/j.1600-0463.2007.apm_630.x ).
- Is the “2.4. String test” really necessary? Since the technique is mainly applied to the genus Klebsiella, and the present study did not have any positive tests. I suggest deleting the test.
- In item 3.1, the antimicrobial resistance data was presented for each of the drugs. It would be more interesting if the approach were separated by antimicrobial class, so that the reader could have a broad view of the MDR classification.
- It is noteworthy that such a complex study of the molecular biology of microorganisms does not emphasize the detection of biofilm formation genes, discussing its findings only with data from the MTP test. What is the justification?
Author Response
Comment 1: The abstract and introduction are adequate. The introduction provides current information on the subject.
RESPONSE 1: We are thankful for the Reviewer’s comment.
Comment 2: Lines 84-85 cite the classic reference, in this case Stepanovic 2007 (10.1111/j.1600-0463.2007.apm_630.x ).
RESPONSE 2: In the revised manuscript, the reference has been corrected.
Comment 3: Is the “2.4. String test” really necessary? Since the technique is mainly applied to the genus Klebsiella, and the present study did not have any positive tests. I suggest deleting the test.
RESPONSE 3: String test is also applied to other species. Additionally, even the fact that out isolates were negative to the specific test is a result. Therefore, we strongly believe that string test should stay.
Comment 4: In item 3.1, the antimicrobial resistance data was presented for each of the drugs. It would be more interesting if the approach were separated by antimicrobial class, so that the reader could have a broad view of the MDR classification.
RESPONSE 4: This could be possible for antimicrobials, like aminoglycosides, fluoroquinolones, etc, while for β-lactams, resistance to penicillins or cephalosporins has diverse significance than resistance to carbapenems. However, in Table S1, resistance of each isolate to different antimicrobials is described analytically. Additionally, as it was calculated 44% of the isolates studied were MDR (non-susceptible to ≥1 agent in >3 antimicrobial categories) (please see lines194-195 in the revised manuscript).
Comment 5: It is noteworthy that such a complex study of the molecular biology of microorganisms does not emphasize the detection of biofilm formation genes, discussing its findings only with data from the MTP test. What is the justification?
RESPONE 5: We are thankful for the Reviewer’s comment. Probably, the best explanation is negligence. Additionally, since we sequenced a limited number of isolates, there was no evidence about specific genes. However, in the revised manuscript, the contribution of specific genes in biofilm formation has been discussed (please see lines 348-358).